# Merging Feed-Forward Sublayers
# for Compressed Transformers

**Neha Verma**                                                                 *nverma7@jhu.edu*
*Johns Hopkins University*

**Kenton Murray**                                                              *kenton@jhu.edu*
*Johns Hopkins University*

**Kevin Duh**                                                                  *kevin@cs.jhu.edu*
*Johns Hopkins University*

**Reviewed on OpenReview:** *https://openreview.net/forum?id=t8iuiH46g0*

## Abstract

Pruning is a prevailing model compression method that identifies and removes unimportant parameters based on various importance metrics. In this work, we instead target redundant parameters via parameter merging, proposing a method that combines Transformer feed-forward sublayers through neuron alignment, merging, and weight tying. We find that this method produces compressed models with performance comparable to their original counterparts while tying more than a third of their feed-forward sublayers, and demonstrates improved performance over a strong, generalized layer pruning baseline. For example, this method enables removing 21% of the total parameters from a vision transformer while maintaining 99% of its original performance on ImageNet. We further show our method composes with QLoRA to further shrink base models before fine-tuning, outperforming an equivalent layer-dropping baseline across downstream tasks. Additionally, we observe high activation similarity between different feed-forward sublayers, offering novel insight into their behavior and contextualizing their surprising mergeability.

## 1 Introduction

Advances in deep learning now heavily rely on large pre-trained models to achieve state-of-the-art performance on various tasks. Consequently, model compression techniques that balance efficiency and performance are increasingly vital for enabling model deployment across diverse inference settings and hardware constraints, in order to support widespread use cases.

Much of the prior work in model compression has built upon pruning, quantization, and distillation techniques (LeCun et al., 1989; Fiesler et al., 1990; Hinton et al., 2015). Prior work on pruning introduces many techniques that identify unimportant regions of model parameters that can be removed without drastically changing model performance. These techniques target individual weights, neurons, or general regions of a model—such as attention heads, parameter chunks, or even entire layers (Voita et al., 2019; Lagunas et al., 2021; Sajjad et al., 2023; Ma et al., 2023). Although unimportant features are often the target of pruning, targeting redundancy has been far less explored, despite frequent evidence of redundancies encoded in deep networks (Kornblith et al., 2019; Nanda et al., 2023; Men et al., 2025). Our work addresses this gap by focusing on redundancy-driven model compression.

In targeting memory reduction, pruning may not be the best framework for targeting redundancy in models; in this work, we investigate *merging* sets of similar parameters as a means of identifying and tying such redundancies. While pruning identifies a subset of features to represent a larger group, merging synthesizes information from all available features, which provides an alternative route to memory reduction. Relatedly,

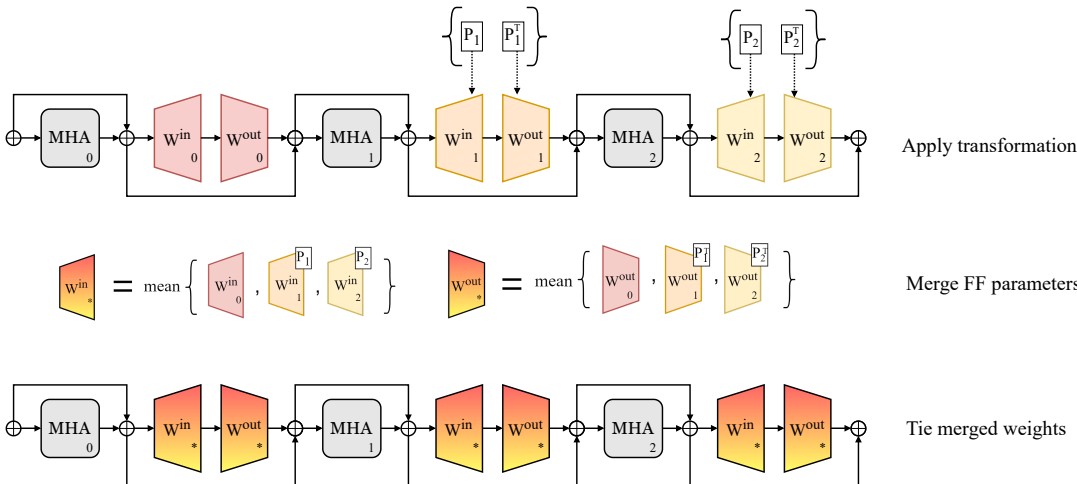

Figure 1: Overview of the feed-forward alignment and merging algorithm used for compression in an example three layers of a Transformer. Multi-headed attention is abbreviated as MHA, feed-forward sublayers are depicted with $W^{\text{in}}$ and $W^{\text{out}}$ weights, and Add&Norm operations are depicted with $\bigoplus$, connected by arrows indicating residual connections.[1] Permutation transformation matrices are shown as $P_i$. Our method includes a permutation finding step, applying the transformations, merging transformed parameters, and finally tying the merged parameters. By merging and tying $k$ feed-forwards, we reduce the model size by $k-1$ feed-forward sublayers.

the field of model merging has explored merging parameters from multiple models to combine their functionalities into one model (Goddard et al., 2024; Yang et al., 2026). We extend ideas from parameter merging to apply to sublayers within one model, rather than just separate models.

To this end, we propose a novel compression method that aligns, merges, and ties separate feed-forward (FF) sublayers within Transformers to achieve a reduced parameter model with reduced memory use (Vaswani et al., 2017). We summarize our approach in Figure 1. We target FF sublayers in particular due to their large overall parameter share, straightforward parameterization, and observed redundancy in their activations. We find that with our method, these groups of FF sublayers are notably compressible via merging. This finding provides insight into their behavior, as well as gives rise to a simple and surprisingly effective method applicable to a variety of existing pre-trained models.

We highlight the contributions of our work:[2]

- We propose and describe a novel parameter merging-based model compression method that selects, aligns, and combines adjacent feed-forward layers in Transformers.
- Across diverse Transformer-based models trained for different applications, including models with gated FFs, we show that merging over one-third of FF sublayers and fine-tuning the resulting model can achieve performance comparable to the original models. We also combine our method with quantization and QLoRA to facilitate even smaller models and smaller fine-tuning settings, respectively (Dettmers et al., 2023).
- We demonstrate that compared to a generalized layer pruning baseline with fine-tuning, our merging method retains more performance across tested models, in both the general and QLoRA settings.
- To explore the surprising effectiveness of merging, we compare FF sublayer outputs from the same model, and find regions with highly similar activations. These same patterns do not occur in attention sublayer outputs.

---

[1]This diagram shows a Post-LN Transformer, but our method easily applies to Pre-LN Transformers as well.
[2]Code is available at `https://github.com/nverma1/merging-ffs-compression`.

Table 1: Positioning of our method against conventional compression approaches for Transformers. FT refers to fine-tuning.

| Method | Granularity | Mem. ↓ | FLOPs ↓ | Training | When preferred |
|---|---|---|---|---|---|
| **Ours** | FF sublayer | ✓ | × | FT | Memory-bound |
| Layer pruning | Full layer | ✓ | ✓ | FT | Latency-bound |
| Unstructured pruning | Single weight | ✓/× | ✓/× | FT | Sparsity-aware hardware available |
| Weight tying at init. | FF / layer | ✓ | × | Pre-train | Designing a new model |

## 2 Related Work

**Weight tying for smaller models**  Prior work has explored numerous methods for employing weight sharing in Transformers, but these methods largely focus on training models from scratch with specific tying schemes. For instance, tying input and output embedding layers in Transformers is a common technique to help cap total parameter count, but more importantly also introduces important gradient sharing for better generalization in language generation tasks (Press & Wolf, 2017; Inan et al., 2017). For non-embedding layers in Transformers, prior work has explored numerous weight tying patterns for pre-training new, efficient models (Dehghani et al., 2019; Lan et al., 2020; Reid et al., 2021; Takase & Kiyono, 2023). Liu et al. (2024) use heavy weight tying between layers at initialization to achieve competitive sub-billion parameter language models. Hay & Wolf (2024) use reinforcement-learning to dynamically select layers to tie while training from scratch in order to reduce the number of active parameters without significantly compromising on performance. Pires et al. (2023) tie widened FF sublayers at initialization and train machine translation (MT) models that outperform standard Transformer MT models. In this work, we extend the usefulness of weight tying for compression in the post-training setting, which remains underexplored.

**Redundancies in Transformers**  Substantial prior work has revealed redundancies within the learned representations of trained Transformers and proposed techniques for reducing them. Dalvi et al. (2020) show high inter-layer similarity in pre-trained Transformer text encoders, and remove redundant neurons using correlation clustering. High inter-layer similarity has also motivated several layer-pruning methods that remove entire model layers followed by optional recovery fine-tuning of the resulting model (Ma et al., 2023; Yang et al., 2024; Men et al., 2025; Gromov et al., 2025). Li et al. (2024) propose a merging-based method for sparsely-activated mixture-of-expert (SMoE) models in order to compress experts; our method extends a similar approach to a wider set of models. Theus et al. (2024) propose to use optimal transport to merge neurons with similar behavior across several layers, demonstrating another recent and novel use of merging for model compression. We position our method in the context of prior work in Table 1.

## 3 Merging Feed-Forward Sublayers

In this section, we motivate Transformer FF sublayers as a merging target, explain permutation-based neuron alignment, and describe our compression method. While we focus on non-gated FFs in this section for simplicity, we also include an extension of our method to SwiGLU FFs, which appear in some of the models used in our experiments, in Appendix A (Shazeer, 2020).

### 3.1 Targeting feed-forward sublayers

We focus our interest on Transformer FF sublayers for several reasons. Firstly, these sublayers account for a majority of non-embedding parameters; they constitute around two-thirds of non-embedding parameters in Transformer encoder or decoder models. Compressing only these parameters still results in substantial overall savings in a model. Secondly, the parameterization of FF sublayers, including common variations, is straightforward, making it a good candidate for merging-based compression approaches that involve parameter alignment.

Beyond these practical considerations, prior work identifies several properties of Transformer FF sublayers that make them well-suited for compression via merging. Li et al. (2023) show that they can be very sparsely activated, where non-zero FF activations can be as low as 3-5%. Other work has claimed that normalization and residual components surrounding FF blocks weaken their effects, which can lead to redundancy in their function (Kobayashi et al., 2024). Finally, Pires et al. (2023) train performant Transformer-based translation models with only one widened and tied encoder FF block, demonstrating useful sharing, but from scratch.

We include a small motivating example demonstrating neuron-level redundancy between adjacent FF layers in a vision transformer (ViT) model (Dosovitskiy et al., 2020). We compute activations following all ViT FF and attention sublayers on over 10,000 image patches from ImageNet-1k (Russakovsky et al., 2015) and compute cross-correlations on activations from adjacent sublayers. For each cross-correlation matrix, we find the neuron pairing that maximizes overall correlation, and then compute the average correlation across the pairing. We then average these values across all layers, and report our results in Table 2. We repeat this process on the same number of randomly sampled data points from $\mathcal{N}(0, \mathbf{I}_d)$, with the same dimensionality as the model activations, and report these results in Table 2 as well. We find that both ViT FFs and attention layers have significantly higher correlation between adjacent layers than random activations, but FFs seemingly encode much more redundancy than attention sublayers.

Table 2: Comparison of average correlations for selected neurons across adjacent ViT feed-forward sublayers, attention sublayers, and randomly generated activations. Feed-forward sublayers have the highest average correlation between adjacent sublayer activations.

| Activations | Average correlation |
|---|---|
| ViT FFs | 0.385 |
| ViT Attn | 0.210 |
| Random Activations | 2.005e-5 |

## 3.2 Background on permutation-based neuron alignment

We propose a merging technique that combines several sublayers into a single parameter set with the goal of enabling effective weight sharing. Our merging technique is inspired by prior work regarding the permutation symmetries of neurons (Li et al., 2015). This technique has been used in studying convergent learning in models and mode connectivity (Tatro et al., 2020; Entezari et al., 2022; Ainsworth et al., 2023).

Permutation-based neuron alignment techniques find an optimal reordering of neurons in one layer that more closely matches ordering of neurons from another layer, without changing the reordered layer's function. Given two layers $\ell_\alpha$ and $\ell_\beta$ to align, we compute only forward passes through both layers using exemplar data to collect activations. This results in two activation sets $X_\alpha$, $X_\beta \in \mathbb{R}^{n \times d}$, where $n$ is the number of example data points, and $d$ is the layer dimension.

To determine corresponding neurons from the activations, we compute cross-correlation $C$, in line with prior work (Li et al., 2015). $\mu$ represents mean vectors, and $\sigma$ standard deviation vectors.

$$C = \text{diag}(\sigma(X_a))^{-1} \mathbb{E}\left[ (X_\alpha - \mu(X_\alpha))^T (X_\beta - \mu(X_\beta)) \right] \text{diag}(\sigma(X_b))^{-1} \tag{1}$$

The resulting matrix $C \in \mathbb{R}^{d \times d}$ reflects how each neuron in $\ell_\alpha$ correlates with each neuron in $\ell_\beta$. To find the neuron alignment that maximizes total correlation, we solve the following optimization problem, where $\Pi_d$ is the set of all permutations of length $d$ (Li et al., 2015; Tatro et al., 2020):

$$\pi^* = \max_{\pi \in \Pi_d} \sum_{j=0}^{d-1} C(j, \pi(j)) \tag{2}$$

This problem is a case of the Linear Assignment Problem, solvable using the Jonker-Volgenant algorithm.[3]

---

[3]We use the implementation provided by `scipy` (Crouse, 2016).

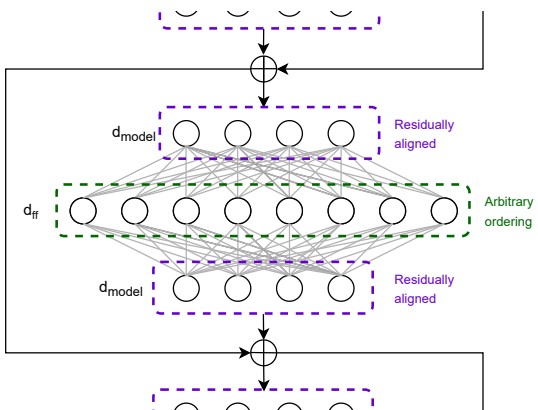

Figure 2: While the input and outputs to Transformer feed-forward layers have inherent ordering via connections to the residual stream, the ordering of the intermediate neurons in these sublayers is arbitrary and the target of our reordering procedure.

## 3.3 Combining feed-forward sublayers

Now, with the appropriate background, we describe our compression method. For our method, we assume we have a predetermined number of FF sublayers $k$ to merge. This number is inferrable given a desired parameter reduction ratio, or set otherwise.

Given a window of $k$ adjacent FF sublayers, we compute forward passes using a small amount of data in order to compute features for each sublayer. In other words, for Transformer FF sublayer $x^{\text{out}} = W^{\text{out}}\phi(W^{\text{in}}x^{\text{in}} + b^{\text{in}})+b^{\text{out}}$, where $W^{\text{in}} \in \mathbb{R}^{d_{\text{ff}} \times d_{\text{model}}}$ and $W^{\text{out}} \in \mathbb{R}^{d_{\text{model}} \times d_{\text{ff}}}$ we obtain features just before the $\phi$ activation. We could additionally consider aligning the neurons just before $W^{\text{in}}$, or just after $W^{\text{out}}$, but prior work has shown that reordering the input to $W^{\text{in}}$ and output of $W^{\text{out}}$ requires permuting many additional weights due to the residual connections in order to maintain functional equivalence (Verma & Elbayad, 2024). Additionally, the residual connections generally impose an inherent ordering on these neurons by design (Elhage et al., 2021). We include a diagram of the specific neurons targeted per layer in Figure 2. For each of the $k$ feed-forward sublayers, we collect $n$ features $X_i \in \mathbb{R}^{n \times d_{\text{ff}}}, \quad i \in [0, k-1]$, where $d_{\text{ff}}$ is the feed-forward dimension.[4]

We designate the first FF sublayer of the set to be an "anchor", and compute the permutation-finding algorithm on each pair of features where one index is always the anchor. In other words, for each sublayer $i \in [1, k-1]$, we find $\pi_i$ between $X_0$ and $X_i$ using the assignment method from Section 3.2.

After converting function $\pi_i$ to its corresponding permutation matrix $P_i$, we transform the $k-1$ non-anchor FF sublayers. We then average these $k$ FF sublayers, and replace each with their average, as in Equations 3–6. Finally, we tie these weights to store just one sublayer, effectively removing the parameters from $k-1$ FF sublayers.

$$W^{\text{in}*} = \frac{1}{k}\left(W_0^{\text{in}} + \sum_{i=1}^{k-1} P_i W_i^{\text{in}}\right) \tag{3}$$

$$b^{\text{in}*} = \frac{1}{k}\left(b_0^{\text{in}} + \sum_{i=1}^{k-1} P_i b_i^{\text{in}}\right) \tag{4}$$

$$W^{\text{out}*} = \frac{1}{k}\left(W_0^{\text{out}} + \sum_{i=1}^{k-1} W_i^{\text{out}} P_i^T\right) \tag{5}$$

$$b^{\text{out}*} = \frac{1}{k}\left(\sum_{i=0}^{k-1} b_i^{\text{out}}\right) \tag{6}$$

---

[4]These layer indices reflect local index within the set of $k$ versus global layer index.

In tying these weights and reducing model size, 1) the model and its gradients occupy less space in memory, making it easier to use smaller hardware and 2) throughput improves due to the reduced memory footprint. Additionally, specific efficient GPU+CPU execution techniques like layer-to-layer that involve on- and off-loading parameters may also benefit from this sharing scheme by reducing the number and size of data transfers (Pudipeddi et al., 2020; Aminabadi et al., 2022).

### 3.4 Selecting sublayers to merge

To identify $k$ adjacent feed-forward sublayers to merge, we employ a sliding window approach. For each starting index $i \in \{0, ..., N_{\text{layers}} - k\}$, we apply the merging procedure outlined in Section 3.3, and evaluate the resulting candidate on a validation set. While we exhaustively test each window, the computational overhead for permutation alignment and parameter arithmetic remains low, scaling only linearly with the number of layers. Additionally, since features and correlations are pre-computed and reused across candidates, the primary bottleneck is limited to testing candidates. The candidate with the highest post-merge evaluation scores is selected, followed by a brief recovery fine-tuning phase to restore performance on the downstream task. We summarize our layer selection method with recovery fine-tuning in Algorithm 1.

---

**Algorithm 1** Feed-Forward Sublayer Merge

---

**Input:** Model parameters $\theta_{\text{in}}$, collected features $\{X_i\}_{i=0}^{N_{\text{layers}}-1}$, batched fine-tuning data $D_{\text{ft}}$
**Input constants:** $k$, $N_{\text{layers}}$, MAXUPDATES
**Initialize:** $\theta_{\text{selected}}$, BESTSCORE $\leftarrow 0$
**for** $i = 0$ **to** $(N_{\text{layers}} - k)$ **do**
    $\theta_{\text{merged}} \leftarrow$ COMPRESS$(\theta_{\text{in}}, \{X_i\}_{i=0}^{N_{\text{layers}}-1}, k)$
    **if** EVAL$(\theta_{\text{merged}}) >$ BESTSCORE **then**
        $\theta_{\text{selected}} \leftarrow \theta_{\text{merged}}$
        BESTSCORE $\leftarrow$ EVAL$(\theta_{\text{merged}})$
    **end if**
**end for**
**for** $i = 0$ **to** MAXUPDATES **do**
    $\theta_{\text{selected}} \leftarrow$ UPDATE$(\theta_{\text{selected}}, D_{\text{ft}}(i))$
**end for**
**Output:** $\theta_{\text{selected}}$

---

## 4 Experimental Setup

To test its extensibility, we apply our compression method to a diverse set of Transformer-based models. Specifically, we use a vision transformer (ViT) (Dosovitskiy et al., 2020), OLMo-1B (Groeneveld et al., 2024), GPT-2 Large (Radford et al., 2019), and a Transformer-based MT model from OPUS-MT (Tiedemann & Thottingal, 2020). We select these models to cover a diversity of Transformer model types (decoder-only, encoder, encoder-decoder) and different modalities. We additionally include experiments on OLMo3-7B (Olmo Team, 2025) using QLoRA (Dettmers et al., 2023).

For each main setting, we list the model used, the example data for computing alignments, and finally the data used for recovery fine-tuning and evaluation. Additional hyperparameters and details are included in Appendix B. More details on dataset statistics can be found in Appendix F.

### 4.1 Image classification with ViT

We use a vision transformer (ViT) for our image classification experiments, with resolution of 224×224, and patch size of 16×16 (Dosovitskiy et al., 2020). ViT is a 12-layer Transformer encoder model pre-trained on ImageNet-21k, and subsequently fine-tuned on ImageNet-1k. ImageNet-1k is a classification task where images belong to one of 1000 categories (Russakovsky et al., 2015). For computing activations, we use around

10,000 patches from the ImageNet-1k validation set. Evaluation results are computed on original validation labels. We fine-tune our ViT models on ImageNet-1k for up to 50,000 steps with a batch size of 128, and report accuracy.

## 4.2 Language modeling

For our experiments, we use OLMo-1B and GPT-2 Large, which are two decoder-only language models trained on extensive English-majority text (Groeneveld et al., 2024; Radford et al., 2019). We note that OLMo-1B has SwiGLU FFs, whereas GPT-2 Large has non-gated FFs. For computing example activations, we use around 10,000 tokens from Wikitext103 validation data (Merity et al., 2017). Finally, we use the train and test splits from Wikitext103 for fine-tuning and evaluation, respectively.

Because we use Wikitext103 for recovery fine-tuning, we also fine-tune the uncompressed GPT-2 and OLMo-1B baseline model for a fair comparison on Wikitext103 evaluation data. Because we use the direct training data to compress our machine translation and ViT models, we do not additionally fine-tune their uncompressed baselines as the fine-tuning data we use already appears in their original training data. We fine-tune both language models for up to 100,000 steps with batches of 2048 tokens for GPT-2 and 4096 for OLMo-1B. We select the best model based on validation perplexity and report test perplexity with a sliding window of 512 tokens.

## 4.3 Machine translation

For our experiments on machine translation, we use a 12-layer Chinese-English Transformer-based translation model from an OPUS-MT release (Tiedemann & Thottingal, 2020). For computing activations, we use around 10,000 tokens from the Tatoeba validation set[5] (Tiedemann, 2020). For fine-tuning, we use original training data released by the Tatoeba translation challenge, sourced from OPUS (Tiedemann, 2012).

We apply our method to both the encoder and decoder separately, constituting two anchors. We test windows in sync, meaning the same window from the encoder and decoder are merged, but separately. We fine-tune our translation models for up to 100,000 steps with a batch size of 64 sentences. We use `sacrebleu` to compute BLEU scores for evaluation (Papineni et al., 2002; Post, 2018), and `pymarian` to compute COMET scores[6] (Rei et al., 2022; Gowda et al., 2024).

## 4.4 Layer pruning baseline

We compare to a generalized, layer-pruning baseline that removes the optimal block of contiguous layers (Ma et al., 2023; Yang et al., 2024; Men et al., 2025; Gromov et al., 2025). We focus on structured pruning as our primary baseline because unstructured methods typically fail to realize memory savings without extreme sparsity ratios and/or specialized sparse libraries. We choose layer pruning as it 1) achieves real-world memory savings and 2) targets compression at the parameter level.

We implement a layer-pruning baseline for comparison that reflects the current state-of-the-art. Many existing methods often rely on various similarity measures to choose adjacent layers to prune. For example, Men et al. (2025) propose a Block Influence measure that computes inner-product based similarity between adjacent layers, and then selects the subset of adjacent layer with the largest overall Block Influence. However, we adopt a more comprehensive sliding window search by pruning and evaluating every potential candidate model on validation data. By selecting the specific set of layers that maximizes compressed performance when dropped, we generalize multiple methods by selecting a compressed model that is at least as performant or better than these heuristic methods. After selecting the best pruned model, we then fine-tune the model with the same specifications as our method. In all, this encapsulates a strong, structured pruning baseline that generalizes many layer-pruning based techniques.

---

[5]Tokens are counted on the source side.
[6]We use the `wmt-22-da` model.

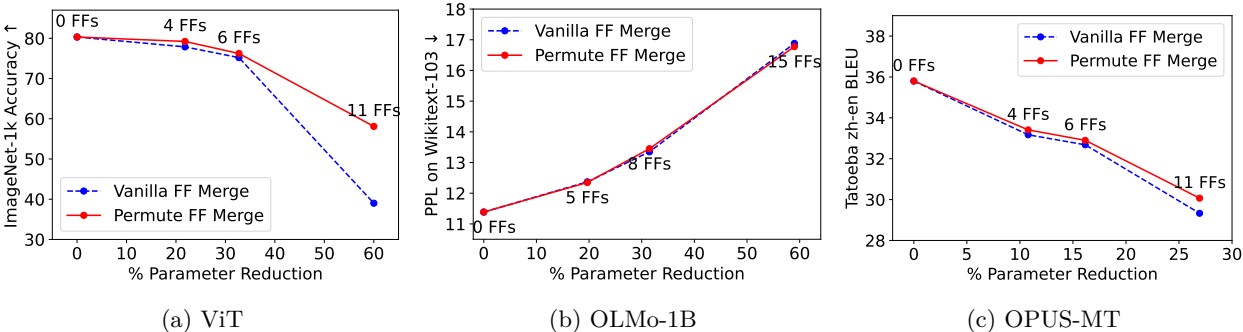

|                    |                     |                    |
| :----------------: | :-----------------: | :----------------: |
| (a) ViT            | (b) OLMo-1B         | (c) OPUS-MT        |

Figure 3: Results across all three tasks depicting compression versus performance results. We include results from our main method, labeled as Permute FF Merge, as well as our method without permutation alignment, depicted as Vanilla FF Merge. We note that our method retains almost complete performance at one-third of feed-forward sublayers removed, across all tasks, and continues to retain high performance at one-half of FF sublayers removed.

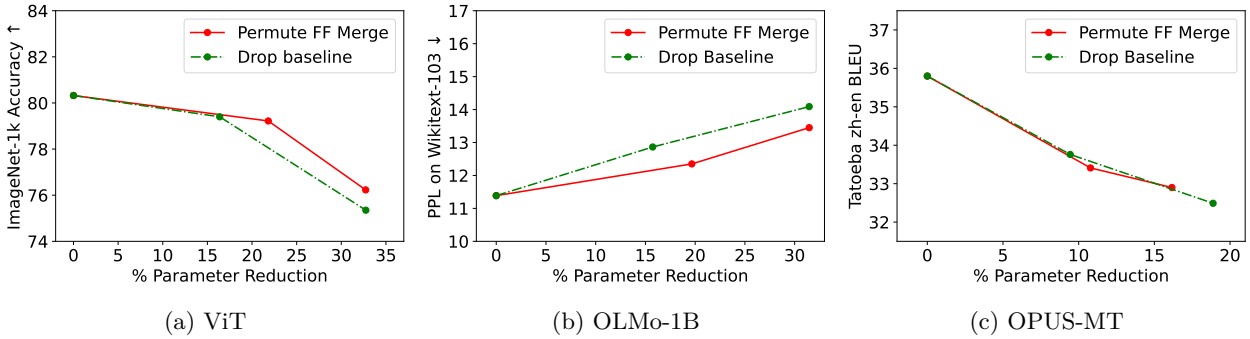

|                    |                     |                    |
| :----------------: | :-----------------: | :----------------: |
| (a) ViT            | (b) OLMo-1B         | (c) OPUS-MT        |

Figure 4: Results across all three tasks depicting compression versus performance for our method and a strong layer-dropping baseline method. We perform layer dropping for 1/6 and 1/3 of layers dropped, and fine-tune the best pre-tuned set of dropped layers for all sliding windows. Across the parameter reduction range shown, our merging-based compression method outperforms or matches layer-dropping across the three tasks.

## 4.5 OLMo3-7B QLoRA Extension

We additionally apply our method alongside 4-bit QLoRA to OLMo3-7B (Olmo Team, 2025) on three downstream task-specific datasets: SamSum (Gliwa et al., 2019), NarrativeQA (Kočiskỳ et al., 2018), and HotPotQA (Yang et al., 2018). SamSum is an English dialogue summarization dataset, NarrativeQA is a story-context question-answering dataset, and HotPotQA is a multi-hop question-answering dataset based on Wikipedia. OLMo3-7B is a 32-layer English language model trained on the open Dolma dataset (Soldaini et al., 2024). We compare our merging method to a layer-drop baseline. We compute features on about 10,000 tokens sampled from Dolma, and select the best pre-tune compressed models using Wikitext-103. We report results on SamSum and NarrativeQA using ROUGE-1, ROUGE-2, and ROUGE-Lsum (Lin, 2004), and report results on HotPotQA using Exact match and F1 scores.

## 5 Results

In discussing compression results, we define compression ratio as the fraction of parameters remaining post-compression compared to the original model, and parameter reduction as the fraction of parameters removed, also compared to the original model. For space reasons, we display language modeling results on OLMo-1B in the main results section, and include additional results on GPT-2 in Appendix C.

## 5.1 Merging feed-forward sublayers across compression ratios

We evaluate our compression method on image classification using ViT, language modeling using OLMo-1B, and machine translation using an OPUS-MT zh-en model, and report our results in Figure 3. Additional results on GPT-2 are in Figure 7a in Appendix C. We report results on 1/3, 1/2 and $(n-1)/n$ removed feed-forward sublayers, to test our method at different overall compression ratios.[7] We also report results from our compression method without the permutation step, denoted as "Vanilla." An average of 3 runs is reported at 1/3 and 1/2 removed feed-forward sublayers, for both the vanilla and permute settings. This setting helps us isolate the effect of parameter alignment in our method.

From our results, we see that even up to 1/2 of FF sublayer parameters removed, which is over 30% in parameter reduction for ViT and OLMo-1B,[8] our method retains high performance compared to the original model. At 1/3 of FF sublayers removed, performance is almost identical to the original model, resulting in only a 1% accuracy drop in ViT, <1 PPL increase in OLMo-1B, and 2 BLEU drop in the translation model. Full numerical results are found in Table 14 in Appendix D.

We also note the effects of our method are consistent across all three of our tasks tested. Given the variety of models tested, including different modalities and Transformer model types, this suggests that our method can generalize to a variety of Transformer-based models. Additionally, we notice that permutation-based compression is routinely better compared to no-permute vanilla baselines, demonstrating the effectiveness of aligning features before merging. This effectiveness is more pronounced at larger numbers of FF sublayers removed. While this effect is diminished in OLMo-1B, it is more present in GPT-2, suggesting some potential added complexity in aligning gated linear units. Despite this, the effectiveness of vanilla and permute-based weight tying is pronounced in both models. In summary, our results show that 1) post-training weight tying is a simple and effective compression method and 2) permutation-based alignment of these shared weights can improve final compression performance.

In Figure 4, we compare our method at 1/3 and 1/2 FFs removed to our layer-pruning baseline. We drop layers to attempt to match the reduction ratios of our own methods, constituting 1/6 and 1/3 of layers dropped for all three models. However, since we cannot match exact ratios due to the granularity of the methods, we plot the exact parameter reduction ratios and performance, and compare. As seen in the figure, as well as in GPT-2 from Figure 7b in Appendix C, our method consistently matches or outperforms the generalized layer-dropping method. This comparison confirms that exploiting the mergeability of feed-forward sublayers is a competitive alternative to strong structured pruning-based methods for model compression. While structured pruning can reduce latency as well as memory usage, in settings where maintaining high performance under memory constraints is important, post-training weight tying can provide a more effective alternative.

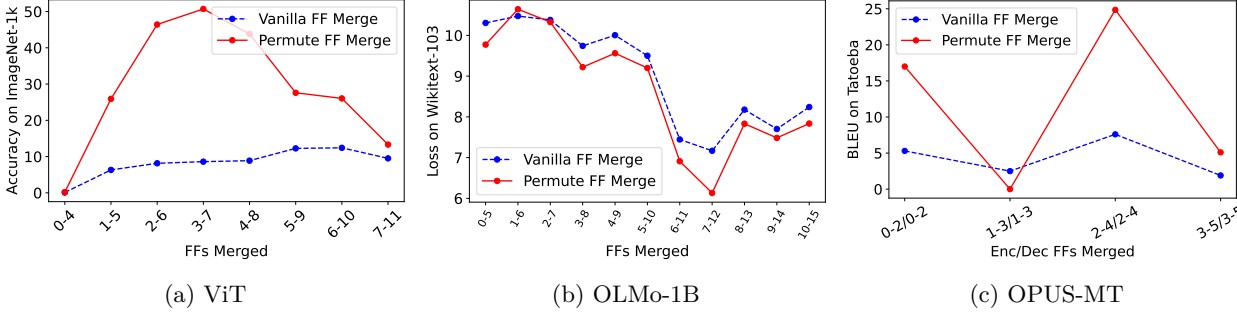

(a) ViT          (b) OLMo-1B          (c) OPUS-MT

Figure 5: Performance curves over different ranges of merged feed-forward sublayers representing 1/3 FFs removed. We display loss on Wikitext-103 for visibility. Across all three tasks, there are clear ranges of merged sublayers that retain more performance when merged.

---

[7]Compression ratios for OPUS-MT differ due to the enc-dec architecture.
[8]We include embedding parameters in all % parameter reduction and compression ratio calculations.

## 5.2 Choice of merged sublayers

In our merging algorithm, we choose which layers to merge by computing performance over sliding windows of $k$ indices. For each of our model/task pairs, we compute the pre-tuning performance of the merging algorithm on 1/3 of FF sublayers dropped across all windows in Figure 5. In this figure, we can see that by simply combining feed-forward sublayers, substantial performance can be retained by our models; for example, after merging half of feed-forwards in both the encoder and decoder of the OPUS-MT model, the model still achieves a BLEU score of $\sim$25, indicating substantial functionality despite the granular, untuned tying of parameters.

Before tuning, it appears that the choice of layers seems to be important, resulting in larger performance variance. However, these differences reduce once recovery fine-tuning is performed. To see this, we randomly select 3 sets of $k$ consecutive layers for each of our tasks, and apply recovery fine-tuning to these compressed models. In Table 3, we observe that models achieve similar performance after fine-tuning. This suggests that more than layer selection, the aligning, tying, and tuning portions of our method drive much of the performance improvement.

Table 3: Results comparing our compression method at 1/3 of feed-forward sublayers removed with different sublayer groups. We include three random consecutive selections of sublayers, excluding the original selection.

|  | ViT Acc.(%) ↑ | OLMo-1B PPL ↓ | GPT-2 PPL ↓ | OPUS-MT BLEU ↑ |
|---|---|---|---|---|
| Best pre-tune | 79.2 | 12.2 | 17.3 | 33.5 |
| Random 1 | 79.5 | 12.2 | 18.3 | 33.9 |
| Random 2 | 78.5 | 12.0 | 17.1 | 33.8 |
| Random 3 | 78.9 | 12.0 | 17.3 | 33.1 |

## 5.3 Choice of anchor layer

We also examine the sensitivity of our method to the choice of anchor layer in our alignment step. In section 3.3, we choose the first FF sublayer in the sequence to serve as the anchor, and compute permutations aligning the remaining sublayers to the anchor. Here, we also consider using either the *last* FF or the *middle* FF of the sequence, and report results in our 1/3 FF merge setting in Table 4.

Given the similar results across settings, our merging approach is robust to the choice of anchor layer, enhancing the reliability of our permutation-based alignment method to find corresponding features for a useful merge.

Table 4: Results comparing our compression method with different anchors at 1/3 of feed-forward sublayers removed.

| Anchor | ViT Acc.(%) ↑ | OLMo-1B PPL ↓ | GPT-2 PPL ↓ | OPUS-MT BLEU ↑ |
|---|---|---|---|---|
| First | 79.2 | 12.2 | 17.3 | 33.5 |
| Middle | 79.5 | 11.9 | 17.4 | 33.4 |
| Last | 79.0 | 12.0 | 17.4 | 33.5 |

## 5.4 Additional compression via quantization

While our method reduces model size via weight tying, quantization also reduces the overall model size via reducing parameter precision. As our method and quantization apply to orthogonal targets, their joint

Table 5: Compression results across three tasks, before and after additional compression via quantization. In this case, compression is measured in terms of total model storage complexity (disk space) instead of parameter count.

| Model | Metric | Our Method | | +LLM.int8() | |
|---|---|---|---|---|---|
| | | Compression | Performance | Compression | Performance |
| ViT | Accuracy(%) ↑ | 78% | 79.21 | 20% | 79.21 |
| OLMo-1B | PPL ↓ | 80% | 12.24 | 48% | 12.25 |
| GPT-2 | PPL ↓ | 80% | 17.27 | 22% | 17.31 |
| OPUS-MT | BLEU ↑ | 89% | 33.54 | 51% | 33.53 |

application may be used together for additional memory savings. We demonstrate with the LLM.int8() method due to its effectiveness and widespread use (Dettmers et al., 2022). Quantization-only results can be found in Appendix G, where we find minimal degradation, if any, in original scores.

We quantize our models with LLM.int8() after removing 1/3 of FF sublayers, and report results in Table 5. Given the minor changes in scores before and after quantization, this suggests these methods indeed do apply orthogonally, and can be used in conjunction. Combining our method with quantization provides even smaller compression ratios, while retaining high performance. This coupling helps to realize compression ratios like 20% when considering total model storage complexity.

### 5.5 QLoRA Extension on OLMo3-7B

In combining our method with QLoRA, we are able to reduce the size of the OLMo3-7B base model further before quantizing and fine-tuning on downstream summarization and QA tasks. We report results on the base model, layer pruning, and our method at 1/3 FFs merged in Table 6.

While our method slightly under-performs the base QLoRA setting, we improve over the Drop baseline setting across datasets and maintain high downstream performance. We note that for both the layer pruning baseline as well as our method, the base model is frozen after compressing, which demonstrates that our method holds up to this aspect of LoRA (Hu et al., 2022), and can aid in settings where base models need to be reduced further before tuning. For both NarrativeQA and HotPotQA, opting for the Merge recovers markedly substantial performance, while still reducing the base model footprint by an additional ∼20%.

Table 6: QLoRA fine-tuning results on SamSum, NarrativeQA, and HotPotQA. We report QLoRA baseline results on the full model, a Layer-drop baseline, and our MergeFF method. EM is Exact Match. Our merging method can help reduce the base model footprint while still allowing for high downstream task performance.

| Method | Layers | SamSum | | | NarrativeQA | | | HotPotQA | |
|---|---|---|---|---|---|---|---|---|---|
| | | ROUGE-1 | ROUGE-2 | ROUGE-Lsum | ROUGE-1 | ROUGE-2 | ROUGE-Lsum | EM | F1 |
| QLoRA | - | 53.83 | 28.99 | 45.17 | 79.63 | 49.81 | 76.95 | 63.74 | 76.82 |
| +Drop | 8-14 | 51.34 | 26.40 | 42.80 | 70.88 | 43.12 | 70.30 | 50.43 | 62.63 |
| +Merge | 18-29 | 52.45 | 27.92 | 44.10 | 77.10 | 46.62 | 76.37 | 63.24 | 76.16 |

### 5.6 Throughput improvements

As previously mentioned, reducing the overall parameterization of a model can lead to throughput improvements in terms of maximum attainable batch size with the reduced model. To demonstrate this effect, we measure the throughput improvements attained with the GPT-2 model and its compressed varieties. Throughput is measured on a consumer-grade GPU (RTX 2080), and with sequences of 128 tokens. To mea-

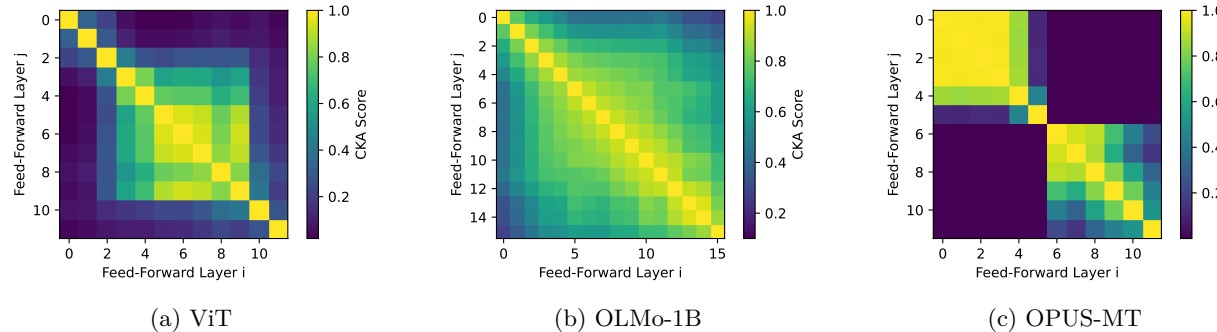

Figure 6: CKA plots of feed-forward sublayer hidden states across three different models. In all three settings, we see clear regions of high similarity between different FF layers.

sure throughput, we find the maximum attainable batch size for either a forward pass (simulating training) or decoding (simulating inference), and report the throughput increase (%) with respect to the uncompressed model. Results are found in Table 7.

As seen in both the improvements for forward passes and decoding, compressing our models via tying FF layers can result in real-world throughput improvements for both training (or fine-tuning) and inference.

Table 7: Throughput improvements (+throughput) on GPT-2 across different levels of compression, for both a forward pass and autoregressive decoding. Due to the model memory savings, throughput on the same hardware notably improves due to more efficient batching (max bsz).

| | Forward Pass | | Decoding | |
| --- | --- | --- | --- | --- |
| Setting | max bsz | +throughput | max bsz | +throughput |
| baseline | 66 | 0.0% | 41 | 0.0% |
| $-1/3$ FFs | 72 | 9.1% | 44 | 7.3% |
| $-1/2$ FFs | 75 | 13.6% | 46 | 12.2% |

### 5.7 Similarity trends across feed-forward sublayers

Given the success of simply aligning and merging adjacent feed-forward sublayers for compression, we look further into possible signs of redundancy in their representations, as alluded to in previous work (Pires et al., 2023; Kobayashi et al., 2024). While earlier we established heightened redundancy in adjacent FF activations in ViT through analyzing cross-correlations, we examine this phenomenon more closely across more models.

To this end, we compare outputs between FF sublayers within the same model. Across our three tasks, we use about 10,000 tokens or patches from task validation sets to compute output states from all feed-forward sublayers. Then, we use Centered Kernel Alignment (CKA) to compute their similarity (Kornblith et al., 2019). We plot CKA values for all pairwise interactions between FF sublayers in three of our model types, shown in Figure 6.

We notice that across all three model/task settings, clear regions of high similarity between FF outputs can be observed, despite FF sublayers being interleaved with multi-headed attention sublayers. We note that similar behavior is not seen in attention sublayers, as demonstrated in Figure 8 in Appendix H. While prior work has shown high similarity between the outputs of adjacent *full* Transformer layers, that result can be explained in part by the residual computations that add prior sublayer outputs to current sublayer outputs (Kornblith et al., 2019; Dalvi et al., 2020). However, in this analysis, we isolate the FF outputs from the stream of residual computations, before this output is added back in, making the observed similarity more

surprising due to the greater independence between FF computations. Any observed similarity is not the product of a clear mediator, like the case of residual connections in full Transformer layers.

## 6 Conclusion

In this work, we propose a novel compression method for Transformer models via merging and tying adjacent sets of FF sublayers. Our method offers an alternative to existing compression techniques and opens new possibilities for parameter merging and tying as effective post-training compression strategies. We demonstrate our method's extensibility across diverse models and tasks, and show that it helps retain high performance even after removing 1/3 of FF sublayers, and outperforms a strong layer pruning baseline. Our method shows strong composition with quantization and QLoRA to achieve even smaller compression ratios, which can help open opportunities for model use and fine-tuning across hardware constraints. Beyond obvious memory and storage savings, we also demonstrate throughput improvements with our compressed models. Finally, we find that several FF sublayers exhibit highly similar activation patterns despite being separated by attention sublayers, which may explain their surprising compressibility via merging.

### Broader Impact Statement

This work offers a path to a new compression paradigm that can help democratize AI and help reduce the environmental costs of AI. At the same time, we acknowledge the dual use nature of this type of work, where improving overall access can make it easier for malicious actors to access and deploy harmful systems.

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

## A  Combining SwiGLU Feed-Forward Layers

The SwiGLU variation of the Transformer FF layer is used across numerous current language models, including OLMo (Groeneveld et al., 2024). The SwiGLU FFN is:

$$\text{FF}_{\text{SwiGLU}} = W^{\text{down}}(\text{Swish}_1(W^{\text{up}}x) \otimes V^{\text{gate}}x) \tag{7}$$

where $\otimes$ is the component-wise product. We exclude biases here for simplicity and lack of inclusion in many real-world models. For more details, we refer the reader to Shazeer (2020). In applying our method to SwiGLU FFs, we highlight three key considerations. Firstly, we set the location of our feature collection as the output of the $W^{\text{up}}$ product, before the activation function. Next, the new merged parameters are computed as the following, where $W^{\text{up}}$ and $V^{\text{gate}}$ share the same permutation:

$$W^{\text{up}*} = \frac{1}{k}\left(W_0^{\text{in}} + \sum_{i=1}^{k-1} P_i W_i^{\text{in}}\right) \tag{8}$$

$$V^{\text{gate}*} = \frac{1}{k}\left(V_0^{\text{gate}} + \sum_{i=1}^{k-1} P_i V_i^{\text{gate}}\right) \tag{9}$$

$$W^{\text{down}*} = \frac{1}{k}\left(W_0^{\text{down}} + \sum_{i=1}^{k-1} W_i^{\text{down}} P_i^T\right) \tag{10}$$

Finally, we note that SwiGLU FF sublayers account for a greater share of total model parameters, meaning that applying our merging method to these sublayers offers even greater parameter savings.

## B  Fine-tuning details

### B.1  GPT-2

Hyperparameter values are found in Table 8.

Table 8: Hyperparameters used for GPT-2 fine-tuning.

| Hyperparameter | Value |
|---|---|
| Start LR | 5e-5 |
| LR Schedule | inv_sqrt |
| fp16 | True |
| batch size | 2 |
| n_steps | 100K |

### B.2 OLMo-1B

Hyperparameter values are found in Table 9. We select the best model with early stopping.

Table 9: Hyperparameters used for OLMo-1B fine-tuning.

| Hyperparameter | Value |
|---|---|
| LR Schedule | cosine |
| LR | 1e-4 |
| Warmup ratio | 0.05 |
| fp16 | True |
| batch size | 2 |
| n_steps | 100K |

### B.3 ViT

Hyperparameter values are found in Table 10.

Table 10: Hyperparameters used for ViT fine-tuning.

| Hyperparameter | Value |
|---|---|
| Start LR | 5e-5 |
| LR Schedule | linear_decay with min |
| decay_steps | 20K |
| Min LR | 1e-6 |
| fp16 | True |
| batch size | 128 |
| n_steps | 50K |

### B.4 Machine Translation

We select our best model using validation BLEU, computed on a 2000 instance subset of the full Tatoeba validation set. Hyperparameter values are found in Table 11.

Table 11: Hyperparameters used for OPUS-MT fine-tuning.

| Hyperparameter | Value |
|---|---|
| Start LR | 5e-5 |
| LR Schedule | inv_sqrt |
| fp16 | True |
| batch size | 64 |
| n_steps | 100K |

### B.5 OLMo3-7B QLoRA

We select our best model using validation loss. Hyperparameter values are found in Table 12. In this setting, we collect activations for alignment after the output of the gating operation, rather than from the output of the non-gate projection.

Table 12: Hyperparameters used for OLMo3-7B QLoRA fine-tuning.

| Hyperparameter | Value |
|---|---|
| Start LR | 5e-4 |
| LR Schedule | `inv_sqrt` |
| warmup ratio | 0.01 |
| fp16 | True |
| batch size | 8 |
| n_steps | 3000 |
| weight decay | 0.01 |
| LoRA rank | 8 |
| LoRA $\alpha$ | 16 |
| LoRA modules | all linear |
| LoRA dropout | 0.2 |

## C   Additional GPT-2 Results

We display additional results for language modeling experiments using GPT-2 in Figure 7, and in Table 13.

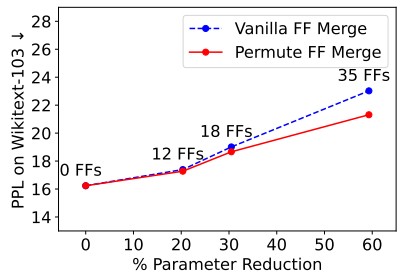
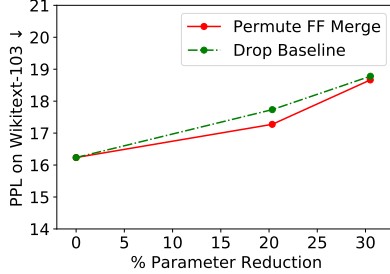
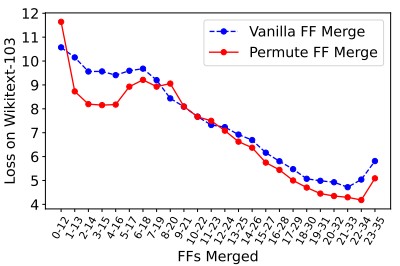

(a) GPT-2 results across compression ratios comparing vanilla and permute methods.

(b) GPT-2 results comparing our FF merge method with permutations to layer pruning.

(c) GPT-2 pre-tuning results with 1/3 FF layers removed.

Figure 7: Additional results on GPT-2 models.

Table 13: Full numerical results on compression results at 1/3 FF sublayers removed, 1/2 FF sublayers removed, and $(n-1)/n$ FF sublayers removed for GPT-2. Original, uncompressed models are included in the first row of results for each model, indicated by 0 FFs removed and no merged indices.

| Model | Metric | Merged Indices | FFs Removed | Vanilla | Permute |
|---|---|---|---|---|---|
| GPT-2 | PPL ↓ | – | 0/36 | 16.16 | 16.16 |
| | | 22-34 | 12/36 | 17.39 | 17.27 |
| | | 16-34 | 18/36 | 19.01 | 18.66 |
| | | 0-35 | 35/36 | 23.02 | 21.31 |

## D   Full Results at varying compression ratios

We report our full results across compression ratios in Table 14.

Table 14: Full numerical results on compression results at 1/3 FF sublayers removed, 1/2 FF sublayers removed, and $(n-1)/n$ FF sublayers removed. Original, uncompressed models are included in the first row of results for each model, indicated by 0 FFs removed and no merged indices.

| Model | Metric | Merged Indices | FFs Removed | Vanilla | Permute |
|---|---|---|---|---|---|
| ViT | Accuracy (%) ↑ | – | 0/12 | 80.3 | 80.3 |
| | | 3-7 | 4/12 | $77.87_{\pm 0.03}$ | $79.22_{\pm 0.02}$ |
| | | 4-10 | 6/12 | $75.20_{\pm 0.06}$ | $76.23_{\pm 0.03}$ |
| | | 0-11 | 11/12 | 39.0 | 58.1 |
| OLMo-1B | PPL ↓ | – | 0/16 | 11.39 | 11.39 |
| | | 7-12 | 5/16 | $12.37_{\pm 0.13}$ | $12.35_{\pm 0.06}$ |
| | | 6-14 | 6/16 | $13.35_{\pm 0.17}$ | $13.45_{\pm 0.15}$ |
| | | 0-15 | 15/16 | 16.88 | 16.77 |
| OPUS-MT | BLEU ↑ | – | 0/12 | 35.8 | 35.8 |
| | | 2-4/2-4 | 4/12 | $33.17_{\pm 0.08}$ | $33.41_{\pm 0.05}$ |
| | | 0-3/0-3 | 6/12 | $32.68_{\pm 0.06}$ | $32.90_{\pm 0.22}$ |
| | | 0-5/0-5 | 11/12 | 29.3 | 30.1 |

# E  COMET evaluation of MT experiments

We report the COMET scores of our main experiments, split by section, in Table 15.

Table 15: Comet scores corresponding to BLEU scores in each table. The first section corresponds to Appendix Table 14, the second Main Paper Table 1, the third Main Paper Table 3, and the last Main Paper Table 2.

| Experiment | Merged Indices | Anchor | FFs Removed | Vanilla | Permute |
|---|---|---|---|---|---|
| Main | – | First | 0/12 | 86.8 | 86.8 |
| | First | 2-4/2-4 | 4/12 | 85.7 | 85.7 |
| | First | 0-3/0-3 | 6/12 | 85.2 | 85.4 |
| | First | 0-5/0-5 | 11/12 | 83.1 | 83.5 |
| Layer choice | First | 0/2-0/2 | 4/12 | - | 85.8 |
| | First | 1-3/1-3 | 4/12 | - | 85.8 |
| | First | 3-5/3-5 | 4/12 | - | 85.6 |
| Anchor choice | Middle | 2-4/2-4 | 4/12 | - | 85.8 |
| | Last | 2-4/2-4 | 4/12 | - | 85.7 |
| +Quantization | First | 2-4/2-4 | 4/12 | - | 85.8 |

# F  Dataset details

We report the dataset statistics for our evaluations and training data used in this work in Table 16. For fine-tuning data, we note that updates reported in per setting in this section give a representation of data usage rather than the training counts provided here.

Table 16: The number of instances used in each fine-tuning and evaluation datasets. Instances are images for ImageNet, lines of text for Wikitext-103, bitext pairs for OPUS/Tatoeba, and dialogue/summary pairs for SamSum.

| Dataset | Train | Validation | Test |
|---|---|---|---|
| ImageNet-1k | 12,281,167 | 50,000 | - |
| Wikitext-103 | 1,801,350 | 3,760 | 4,358 |
| OPUS/Tatoeba | 41,649,946 | 43,074 | 10,389 |
| SamSum | 14,732 | 818 | 819 |

## G Quantization-only results

Table 17: Quantization results across four tasks, before and after LLM.int8() quantization. In this case, compression is measured in terms of total model storage complexity (disk space) instead of parameter count.

| Model | Metric | Base Model | +LLM.int8() | |
|---|---|---|---|---|
| | | Performance | Compression | Performance |
| ViT | Accuracy(%) ↑ | 80.30 | 25% | 80.30 |
| OLMo-1B | PPL ↓ | 11.39 | 58% | 11.40 |
| GPT-2 | PPL ↓ | 16.16 | 28% | 16.17 |
| OPUS-MT | BLEU ↑ | 35.80 | 57% | 35.60 |

## H Attention sublayer similarity

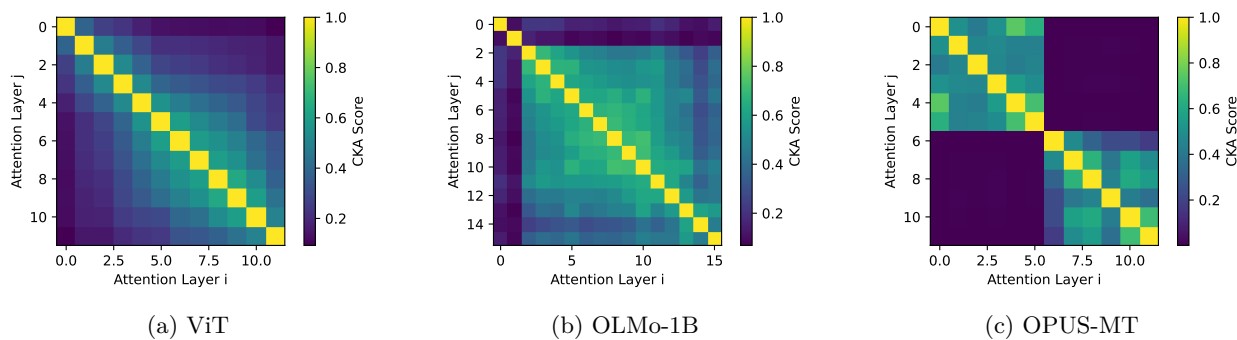

(a) ViT

(b) OLMo-1B

(c) OPUS-MT

Figure 8: CKA plots of multi-headed self-attention sublayer activations across three different trained models. Attention activations are largely dissimilar from each other across model types. We do not compare between encoder and decoder attention sublayers in the translation model due the differences in token inputs.

