# OpenReview forum: "Merging Feed-Forward Sublayer for Compressed Transformers"
_TMLR — Accepted by TMLR_

### Review · Reviewer_13xY · 2026-04-13

**Summary Of Contributions:**

Summary
- The paper proposes a pruning method that merges consecutive feed-forward layers by identifying redundant layers, finding permutation matrices that maximize cross-correlation via a linear assignment problem, and then averaging the aligned weight matrices.
- To determine which consecutive feed-forward sublayers to merge, the authors exhaustively evaluate candidate subsets in a sliding-window manner. After merging, they further fine-tune the model to recover performance.
- The method is evaluated on ViT classification tasks, perplexity on OLMo, and BLEU on a machine translation model, and the results are shown to be broadly comparable to naive baselines.

**Audience:**

Yes

**Audience Explanation:**

While the paper is clearly written, its empirical support appears limited, for the reasons detailed above.

**Broader Impact Concerns:**

The paper includes a broader impact statement.

**Claims And Evidence:**

No

**Claims Explanation:**

- My main concern is that the permutation-based merging, which is the paper’s main contribution, provides only marginal improvements over vanilla merging for generative models, as shown in Figures (b) and (c). The method appears to achieve competitive speedups primarily for ViT on classification tasks, which limits its broader applicability.
- Moreover, the performance degradation in generative models is nontrivial overall. Even with only a 10-20% reduction in parameters, there appears to be a substantial drop in performance, as reflected in both perplexity and BLEU.
- The evaluation also lacks stronger downstream benchmarks, such as advanced QA and reasoning tasks on more recent LLMs.

**Requested Changes:**

- Can you add evaluation results on stronger downstream benchmarks for LLMs, such as advanced QA or reasoning tasks, on more recent modern LLMs?

---

> ### Author Response · Authors · 2026-05-08
> **Response to reviewer 13xY**
>
> Thanks for your detailed read and assessment of our work. We would like to respond to your requested changes and weaknesses from your assessment.
>
> **Requested changes**:
>
> > Can you add evaluation results on stronger downstream benchmarks for LLMs, such as advanced QA or reasoning tasks, on more recent modern LLMs?
>
> We have added results below for the newer OLMo3-7B on new QA benchmarks (NarrativeQA & TriviaQA) and on SamSum, which is the same as the OLMo-7B experiment in the paper. The results now include GPT2 (2019), OLMo-1b and 7b (2024), and OLMo3-7B **(Oct 2025)**. Following the QLoRA settings in Section 5.5, we apply our merging method (Permute 18-29, removing 11 FFs out of 32) and a layer-pruning baseline (Drop 8-14, dropping 7 layers) at comparable compression ratios. Results appear below:
>
> **HotpotQA (Exact Match / F1)**
> | Config | EM | F1 |
> |------------------|-------|-------|
> | Baseline | 63.74 | 76.82 |
> | Permute 18-29 | 63.24 | 76.16 |
> | Drop 8-14 | 50.43 | 62.63 |
>
> **NarrativeQA (ROUGE-1 / ROUGE-2 / ROUGE-L)**
> | Config | R1 | R2 | RL |
> |------------------|-------|-------|-------|
> | Baseline | 79.63 | 49.81 | 78.95 |
> | Permute 18-29 | 77.10 | 46.62 | 76.37 |
> | Drop 8-14 | 70.88 | 43.12 | 70.30 |
>
> **SamSum (ROUGE-1 / ROUGE-2 / ROUGE-L)** (on OLMo3-7B, not the same as in the paper)
> | Config | R1 | R2 | RL |
> |------------------|-------|-------|-------|
> | Baseline | 53.83 | 28.99 | 45.17 |
> | Permute 18-29 | 52.45 | 27.92 | 44.10 |
> | Drop 8-14 | 51.34 | 26.40 | 42.80 |
>
> These results strengthen the empirical argument in our paper. Our merging method retains near-baseline performance on these QA benchmarks. On HotpotQA, the merged model retains almost all baseline EM/F1, indicating that even multi-hop reasoning capability is largely preserved through compression. On NarrativeQA and SamSum, the merged model again retains almost all of the baseline performance across all ROUGE metrics. The Drop setting does not see the same amount of performance recovered; in a memory-bound setting, our method can help retain more performance than similar structured pruning methods. We are adding these new results to our paper, replacing those present for OLMo-7B to reflect newer models and harder evaluations as per your requested change.
>
> We would like to also briefly add that our paper's scope is intentionally broader than LLMs, as we also include results on Vision Transformers (encoder) and traditional translation models (encoder-decoder). The choice to test across modeling architectures was made to demonstrate that FF redundancy is a general property of Transformers rather than an LLM-specific phenomenon. However, with these new findresultsngs, we believe that the LLM applications are now more compelling.
>
> **Weaknesses**:
>
> > the permutation-based merging, which is the paper’s main contribution, provides only marginal improvements over vanilla merging for generative models,
>
> We appreciate this observation and would like to clarify both the scope of our contribution and the empirical picture across generative models. First, we view the contribution of the paper as the broader framework of aligning, merging, and tying adjacent FF sublayers as a redundancy-driven compression axis. Permutation alignment is one component of this framework rather than the whole of it. The vanilla-merge results themselves represent a substantive finding: that FF sublayers are mergeable at all without catastrophic loss is, to our knowledge, novel. Even if the gain from permutation alignment is smaller than the gain from only merging and tying, it comprises a portion of our method and still shows improvements, justifying its contribution within the proposed method.
>
> We thank the reviewer for their careful reading and the suggestion to extend our LLM evaluation, which has strengthened the empirical claims of the paper.

---

### Review · Reviewer_36rr · 2026-04-17

**Summary Of Contributions:**

The paper presents a simple post training compression idea for Transformers by focusing on redundancy in the feed forward layers. It aligns neurons across layers, merges them by averaging, and then shares the same weights across nearby layers to save memory without changing the model structure or computation. The method is tested on different types of models including vision transformers, language models like OLMo and GPT 2 Large, and a machine translation model, and it shows that even after tying a large portion of feed forward layers the performance stays close to the original and often does better than a strong pruning baseline. The authors also study the activations using CKA and find that these layers are quite similar to each other, which helps explain why this approach works.

## Strength
1. The method is technically well grounded. Aligning neurons before merging is important since FFN neurons do not have a fixed order, so simple averaging would likely break the representation.
2. The empirical evaluation is quite broad. The method works across encoder, decoder, and encoder decoder models and on different tasks (ImageNet, WikiText, and Machine Translation), which suggests the redundancy is a general property of Transformers.
3. The CKA analysis is a strong part of the paper. It shows that FF layers are much more similar to each other than attention layers, which supports the main idea.
4. The method can be combined with techniques like quantization and QLoRA. This makes it practical for real use cases with limited VRAM.

## Weaknesses

1. The method reduces memory but not computation. Since tied layers are still executed multiple times, there is no speedup, which limits usefulness in latency critical settings.
2. The alignment step may not scale well. It relies on a cubic complexity algorithm, and the paper does not provide enough evidence of efficiency for very large models.
3. The approach depends heavily on recovery fine tuning. Large training budgets are needed, making it unclear how much gain comes from the method itself versus re optimization.
4. There are no ablations on fine tuning budget or data size. This makes it hard to understand how sensitive the method is to training resources.
5. The baseline comparisons are limited. The paper does not include recent redundancy aware or neuron merging approaches, which are closely related.
6. The connection to similar work in other domains like speech or multimodal models is weak, so the positioning feels incomplete.
7. The window selection process relies on exhaustive validation search, which could be expensive and may not scale well.
8. The method assumes the validation set is representative, but this dependency is not analyzed or discussed.

**Audience:**

Yes

**Audience Explanation:**

A lot of TMLR readers would be interested because the paper offers a practical way to reduce Transformer memory usage without significantly hurting performance, which is important for training and fine-tuning large models under limited GPU resources. It also provides useful insight into redundancy in feed-forward layers, showing that many layers learn very similar functions and can be merged, which is relevant for model efficiency and understanding deep representations.

**Claims And Evidence:**

Yes

**Claims Explanation:**

Overall, the paper makes a convincing case for its main idea. The authors test the method on different types of models, such as vision, language, and translation. This makes the claims seem more reliable. They also go beyond pure performance numbers and try to explain why the method works. The CKA analysis is a strong part here, as it clearly shows that many feedforward layers are doing very similar things, giving solid evidence of redundancy before applying the merging. The comparisons with layer pruning are also useful, showing that merging tends to preserve performance better than simply removing layers. On top of that, the method works well with practical techniques like quantization, which adds to its real world relevance. That said, a few gaps remain. It is still not fully clear how much of the benefit comes specifically from the alignment step, since there is no comparison with a simple random tying baseline. The paper does show clear memory savings, but it doesn't talk much about speed, which is important for a lot of applications.

**Requested Changes:**

See weaknesses

---

> ### Author Response · Authors · 2026-05-09
> **Response to reviewer 36rr**
>
> Thanks for your detailed read and assessment of our work. We appreciate your assessment of the strengths of the paper as it captures the key contributions we wished to convey with our work. We respond to the weaknesses you raised below.
>
> > The method reduces memory but not computation.
>
> This is correct. We explicitly frame our goal as reducing memory, acknowledge that reduced memory can indirectly improve throughput (Section 5.6), but do not target direct latency improvements in this work. This is consistent with prior work targeting low-memory settings via layer sharing, such as MobileLLM, Subformer, and EdgeFormer.
>
> > The alignment step may not scale well.
>
> This is a fair concern. The Linear Assignment Problem we solve is O(d_ff^3) per merge, which is non-negligible at scale (e.g., d_ff = 11,008 for OLMo-7B in our paper). However, two factors mitigate the overall cost here: (1) the alignment is a one-time pre-processing step that does not affect inference, and (2) features and correlations are precomputed once and reused across all sliding-window candidates (Section 3.4). For substantially larger d_ff, approximate assignment algorithms (e.g., Sinkhorn-based) could be a natural drop-in replacement, though we have not needed them at the scales tested.
>
> > Limited baseline comparisons; no recent redundancy-aware or neuron-merging approaches.
>
> Thank you for raising this. We would like to clarify the design of our baseline. Rather than implementing one specific layer-pruning method, we adopt an exhaustive sliding-window search that selects the best candidate before recovery fine-tuning, which by construction generalizes (and matches or exceeds) the heuristic-based methods it subsumes such as Block Influence (Men et al., 2024) and ShortGPT-style similarity selection. Regarding neuron-merging approaches specifically, we have added a more explicit discussion of how our method differs to the Related Work section.
>
> We have also included new experiments in response to reviewers QLxK and 13xY, including additional models and benchmarks, and variance reporting. We encourage you to review those new experiments as well if you were interested in determining what has been added in the author response period. We have also updated the PDF with several writing improvements. Thank you for your careful reading of this work!

---

### Review · Reviewer_QLxK · 2026-04-26

**Summary Of Contributions:**

This paper proposes an alternative approach to conventional deep neural network compression by introducing a method that integrates and merges Feed-Forward (FF) sublayers within Transformer architectures rather than relying on traditional parameter pruning. By identifying and exploiting the inherent internal redundancy among these sublayers, the authors demonstrate that multiple FF components can be merged through neuron alignment and weight tying to significantly reduce the total parameter count while maintaining model performance. This process involves aligning weights across layers, merging redundant sublayers, and subsequently fine-tuning the model, with the added capability of being combined with quantization to further minimize memory and computational costs. Extensive empirical evaluations across diverse tasks, including image recognition with Vision Transformers (ViT), language modeling with OLMo-1B, and machine translation with OPUS-MT, reveal that the proposed approach exhibits a good trade-off between the model size and the performance. The results show that this approach consistently outperforms a conventional layer-dropping technique, and the authors substantiate their claims regarding the effectiveness of sublayer merging through detailed observations of high inter-layer similarity and activation patterns.

# Strength

- The method enables significant parameter reduction in existing pre-trained Transformer models by aligning, merging, and sharing sublayer weights, leveraging high inter-layer correlations without requiring training from scratch.

- The approach is substantiated by experimental evidence demonstrating strong similarity between layers, providing a clear and logical basis for sublayer merging.

- A major advantage is its modularity; the technique can be seamlessly combined with other compression methods, such as quantization, layer dropping, and conventional weight-sharing strategies.

- The proposed framework exhibits versatility across various model architectures and diverse tasks, proving its effectiveness in a wide range of practical scenarios.

# Weakness

- The evaluation relies on only a single baseline method, which is inadequate for a comprehensive assessment. It remains unclear how the proposed approach compares to a broader range of established post-hoc compression techniques, such as various magnitude pruning or low-rank approximation methods.

- The experiments appear to have been conducted only once for each configuration, with no reporting of performance variance or error bars. Consequently, it is impossible to determine whether the observed performance differences are statistically significant or merely the result of experimental noise.

**Additional Comments:**

In Algorithm 1, the variable BestScore is never updated.

At the end of the paragraph immediately preceding Section 4.1, is "can be found in F" a typo for "can be found in Appendix F"?

In the final sentence of Section 4.2, I could not understand what "We select the best model based on validation" refers to. It is unclear why multiple models exist in this context; specifically, it should be clarified what the selection is the "best among."

In the comparisons shown in Figure 4, results are only provided for a relatively low parameter reduction ratio. In contrast, Figure 3 evaluates ratios exceeding 50%. It would be desirable to evaluate the comparisons in Figure 4 across a similarly broad range.

**Audience:**

Yes

**Audience Explanation:**

The paper proposes a novel approach to model compression by focusing on the inherent redundancy within Transformer architectures. Given that this method can be integrated with existing techniques—such as quantization—it represents an interesting piece of research that provides a foundation for further investigation.

**Broader Impact Concerns:**

no concern

**Claims And Evidence:**

No

**Claims Explanation:**

While the submission provides empirical evidence demonstrating that merging Feed-Forward (FF) sublayers is a viable approach for model compression, the evidence supporting the claims of its superiority over existing methods is currently incomplete and not entirely convincing.

Specifically, although the authors state that their approach maintains performance better than traditional pruning methods, they rely on a single baseline for comparison. This limited selection raises concerns regarding the baseline's adequacy and fails to provide a comprehensive view of how the proposed method compares to a wider array of established compression techniques. Furthermore, since the experiments were conducted only once for each configuration, it is impossible to determine whether the reported performance gains represent a statistically significant improvement or are simply the result of variance.

Moreover, the paper lacks a critical analysis of the performance-to-model-size tradeoff for standalone quantization. While Table 4 indicates that combining the proposed merging method with quantization effectively reduces model size while preserving performance, the absence of a "quantization-only" baseline makes it difficult to determine the actual added value of the proposed method. Without this comparison, it remains unclear whether the reported results are a unique benefit of the sublayer merging or if similar efficiency could be achieved through standard quantization alone. Consequently, the relative strengths of the proposed approach compared to conventional research are not yet sufficiently substantiated.

**Requested Changes:**

Additionally, the following points must be addressed.

In the Related Work section, although conventional methods are listed, their positioning relative to the proposed approach is not entirely clear. It would be beneficial to include a table summarizing the characteristics and positioning of each method, clarifying in which situations the proposed approach is preferable and in which situations conventional methods are required. Additionally, the content described in Section 4.4 should have been introduced within the Related Work section.

Equation (1) is mathematically incorrect. $\sigma$ must be defined as a diagonal matrix. $C$ cannot be correctly defined unless $1/\sigma$ is treated as an inverse matrix ($\Sigma^{-1}$).

---

> ### Author Response · Authors · 2026-05-08
> **Response to Reviewer QLxK**
>
> Thank you for your detailed read and assessment of our work. We respond to your requested changes, weaknesses, and additional comments below.
>
> **Requested Changes:**
>
> > include a table summarizing the characteristics and positioning of each method, clarifying in which situations the proposed
> approach is preferable and in which situations conventional methods are required.
>
> Thanks for the constructive suggestion. We have added this table to the Related Work section in the updated PDF. We hope this table helps better place our method in context of related methods.
>
> > The content described in Section 4.4 should have been introduced within the Related work section
>
> Thank you for this suggestion. We would like to gently note that layer pruning (the subject of Section 4.4) is in fact already covered in our Related Work section, under the "Redundancies in Transformers” paragraph. To make this clearer in the updated PDF, we have moved a portion of the description of these methods from Section 4.4 to where they are first introduced in Related Work, so that the reader encounters more of the relevant context earlier.
>
> > Equation 1 is mathematically incorrect
>
> Thank you for catching this. You are right that the notation was inconsistent; the standard deviations were not converted to diagonal matrices as they should have been. We have corrected this in the updated PDF.
>
> **Weaknesses:**
>
> > The evaluation relies on only a single baseline method [...] unclear how the proposed approach compares to a broader range of established post-hoc compression techniques
>
> Thank you for raising this point, we would like to clarify this baseline and why we believe our evaluation is more comprehensive than it may initially appear. Our layer-pruning baseline is a generalization of a family of related methods (as described in Section 4.4); rather than implementing one specific layer pruning approach, we adopt an exhaustive sliding-window search that selects the best candidate before recovery fine-tuning. These family of methods instead select contiguous layers based on heuristic similarity measures. By construction, this baseline performs at least as well as the heuristic-based methods it generalizes (e.g., Block Influence, ShortGPT-style similarity selection cited in the paper), making it a strengthened, generalized baseline.
>
> Regarding magnitude pruning, we deliberately focus on structured pruning methods because unstructured pruning (like weight magnitude pruning) does not yield real memory savings without specialized memory formats. Rather than positioning our method as a replacement for all post-hoc compression techniques, we would like to convey it as a new redundancy-driven axis that composes with existing methods.
>
> > the paper lacks a critical analysis of the performance-to-model-size tradeoff for standalone quantization [...] the absence of a "quantization-only" baseline makes it difficult to determine the actual added value of the proposed method.
>
> We appreciate this concern and would like to respectfully push back, as we believe it reflects a different framing of Table 4 (+quantization) than we intend. Quantization compresses models by reducing parameter precision; our method compresses models by exploiting structural redundancy across feed-forward sublayers. Because these target fundamentally different sources of redundancy, they compose without interference, and the purpose of Table 4 is to establish exactly this orthogonality. When LLM.int8() is applied to a model that has already had 1/3 of its FF sublayers merged, we observe essentially no additional performance degradation beyond what merging alone produces. This near-zero interference supports our argument of orthogonal compression axes and is the comparison Table 4 is designed to make.
>
> A quantization-only baseline would frame our method as a competitor to quantization rather than as composable with it, which we believe is your concern as well. We do not believe this is the right framing in this work. Pruning, distillation, low-rank methods, weight tying, and other structural methods are routinely evaluated without being benchmarked against quantization alone, because they operate on a different axis. It is the case that many quantization methods could outperform many of these structural methods. However, their exploration is important as well because quantization has hard precision floors below which performance collapses. Structural methods (like ours) offer a complementary axis that extends the achievable compression frontier beyond what quantization can reach in isolation, as demonstrated in Table 4.

---

> ### Author Response · Authors · 2026-05-08
> **Response continued**
>
> > experiments appear to have been conducted only once for each configuration
>
> Thank you for this fair point. We agree that additional runs per experiment strengthens empirical claims. We would like to highlight robustness evidence already present in the paper, and then describe additional experiments. Although they are not framed as variance estimates, some our ablations serve this purpose. Table 2 shows consistency across fine-tuning experiments; the within-model spread is small relative to the gaps we report against the layer-pruning baseline. Again, with varying anchors in Table 3, different runs produce tightly clustered results across all models.
>
> We have run 2 additional fine-tuning seeds for the main configurations on ViT, OLMo-1B, and OPUS-MT, giving n=3 seeds per configuration when combined with the original seed results from Table 12. The permutation-finding step itself is deterministic, so variance arises only from recovery fine-tuning. Mean ± standard error results are shown below. We are working on incorporating these variance results into our PDF.
>
> | Model (metric) | Setting | Vanilla | Permute |
> |------------------|------------------|------------------|----------------------|
> | ViT (Acc %, ↑) | 3-7 (4/12 FFs) | 77.87 ± 0.03 | 79.22 ± 0.02 |
> | ViT (Acc %, ↑) | 4-10 (6/12 FFs) | 75.20 ± 0.06 | 76.23 ± 0.03 |
> | OPUS-MT (BLEU, ↑)| 2-4 (4/12 FFs) | 33.17 ± 0.08 | 33.41 ± 0.05 |
> | OPUS-MT (BLEU, ↑)| 0-3 (6/12 FFs) | 32.68 ± 0.06 | 32.90 ± 0.22 |
> | OLMo-1B (PPL, ↓) | 7-12 (5/16 FFs) | 12.37 ± 0.13 | 12.35 ± 0.06 |
> | OLMo-1B (PPL, ↓) | 6-14 (8/16 FFs) | 13.35 ± 0.17 | 13.45 ± 0.15 |
>
>
> These results support the claims made in the paper:
> 1. Variance is small across all settings, which hopefully helps the reliable reporting of results from our work.
> 2. Permutation alignment provides mean improvements over vanilla merging on ViT and OPUS-MT. The gains are clear and exceed combined standard errors on ViT (both settings) and OPUS-MT 2-4.
> 3. On OLMo-1B, vanilla and permute perform comparably in mean. The two methods produce similar results within seed variation at both 5/16 and 8/16 settings. We view this as consistent with our discussion in Section 5.1: OLMo-1B uses SwiGLU FFs, and we hypothesize that aligning gated linear units is more challenging than aligning standard FFs. This is also consistent with the GPT-2 results in Appendix Figure 7a (a non-gated decoder-only model), where the permutation advantage is more pronounced, especially at higher compression ratios. The OLMo result therefore reflects a model-architecture-specific limitation of the alignment step rather than a general absence of effect on generative models.
>
> We will incorporate these results into the final paper.
>
> **Additional comments:**
>
> We have corrected the two typos mentioned in the PDF.
>
> > it should be clarified what the selection is the "best among."
>
> Section 4.2 describes the sliding window layer selection method where the best contiguous set of layers is selected for fine-tuning.
>
> > It would be desirable to evaluate the comparisons in Figure 4 across a similarly broad range.
>
> Thank you for the suggestion. Figures 3 and 4 play intentionally different roles: Figure 3 characterizes the full operating range of our method (including aggressive compression), while Figure 4 compares against layer pruning at the more realistic compression ratios for which layer-pruning baselines were originally designed and evaluated. Extending Figure 4 to >50% reduction would compare against operating points where the layer-pruning baseline degrades sharply for reasons unrelated to our method, making the comparison less informative.
>
> _____
>
> We have also added new QLoRA experiments on a newer LLM (OLMo3-7B), across 2 new QA tasks and the same SamSum summarization task, in response to Reviewer 13xY. Please refer to the appropriate author response if you were interested in determining what has been added in the author response period.
>
> We thank the reviewer for the detailed review with many constructive criticisms. Many of these suggestions, like the positioning table in Related Work, explicit variance reporting, the notational fix in Equation 1, and the integration of Section 4.4 material into Related Work, have all led to concrete revisions that are incorporated into the revised paper. On clarification points, we have tried to lay out our reasoning clearly and are happy to continue the discussion if our justifications are not sufficient. We appreciate your careful engagement with our work.

---

> > ### Comment · Reviewer_QLxK · 2026-05-09
> >
> > Thank you for your response.
> >
> > > We appreciate this concern and would like to respectfully push back, as we believe it reflects a different framing of Table 4 (+quantization) than we intend. Quantization compresses models by reducing parameter precision; our method compresses models by exploiting structural redundancy across feed-forward sublayers. Because these target fundamentally different sources of redundancy, they compose without interference, and the purpose of Table 4 is to establish exactly this orthogonality. When LLM.int8() is applied to a model that has already had 1/3 of its FF sublayers merged, we observe essentially no additional performance degradation beyond what merging alone produces. This near-zero interference supports our argument of orthogonal compression axes and is the comparison Table 4 is designed to make.
> > > A quantization-only baseline would frame our method as a competitor to quantization rather than as composable with it, which we believe is your concern as well. We do not believe this is the right framing in this work. Pruning, distillation, low-rank methods, weight tying, and other structural methods are routinely evaluated without being benchmarked against quantization alone, because they operate on a different axis. It is the case that many quantization methods could outperform many of these structural methods. However, their exploration is important as well because quantization has hard precision floors below which performance collapses. Structural methods (like ours) offer a complementary axis that extends the achievable compression frontier beyond what quantization can reach in isolation, as demonstrated in Table 4.
> >
> > While I understand the authors' perspective, the original comment stems from a fundamental question: whether quantization alone could achieve a comparable performance-to-size trade-off. If the proposed method indeed provides an orthogonal benefit to quantization, it should be clearly demonstrated that the combination (Proposed + Quantization) outperforms both the proposed method in isolation and a standalone quantization baseline.
> > Furthermore, the concluding statement that "structural methods [...] extend the achievable compression frontier beyond what quantization can reach in isolation" does not appear to be clearly supported by Table 4 (or Table 5). As it stands, the results only seem to show that the combination is superior to the proposed method alone, rather than establishing a clear advantage over quantization in isolation.
> >
> > > We have run 2 additional fine-tuning seeds for the main configurations on ViT, OLMo-1B, and OPUS-MT, giving n=3 seeds per configuration when combined with the original seed results from Table 12. The permutation-finding step itself is deterministic, so variance arises only from recovery fine-tuning. Mean ± standard error results are shown below. We are working on incorporating these variance results into our PDF.
> >
> > Regarding the additional seeds, increasing to three trials significantly improves the empirical validity compared to a single-run evaluation. I note that these results have not yet been reflected in the manuscript, and I look forward to seeing them fully incorporated in the revised version.

---

> ### Author Response · Authors · 2026-05-09
> **Response to Reviewer QLxK response**
>
> Thank you for your prompt response and continuing to engage with our work.
>
> **Quantization**:
>
> I understand your point, and thanks for understanding our perspective on our framing. Even though quantization is a strong and orthogonal compression method, knowing how much compression is achieved with just quantization is important to make the claim: "structural methods [...] extend the achievable compression frontier beyond what quantization can reach in isolation." I believe this is what you are communicating in your response.
>
> From the following results, which display compression ratio/performance, we can see that the compression ratios achieved by Ours + Quant are not achievable with either method alone, demonstrating that we are indeed extending the achievable compression frontier beyond what quantization or our method can reach in isolation. The following is an extension of Table 5 in the paper.
>
> | Model    | Metric    | Original       | Ours alone     | Quant only       | Ours + Quant   |
> |----------|-----------|----------------|----------------|------------------|----------------|
> | ViT      | Acc(%) ↑  | 100% / 80.3   | 78% / 79.21    | 25% / 80.3    | 20% / 79.21    |
> | OLMo-1B  | PPL ↓     | 100% / 11.39   | 80% / 12.24    | 58% / 11.40   | 48% / 12.25    |
> | GPT-2    | PPL ↓     | 100% / 16.16   | 80% / 17.27    | 28% / 16.17   | 22% / 17.31    |
> | OPUS-MT  | BLEU ↑    | 100% / 35.80   | 89% / 33.54    | 57% / 35.6   | 51% / 33.53    |
>
>
> **Updating the PDF**:
> We will add this information to the final paper. Regarding adding this new information and additional seed results to the final manuscript, we agree that these additions warrant inclusion and assure you we intend to include them. We are actively working on a comprehensive revision that incorporates the multi-seed variance analysis, the quantization-only baselines, and the additional downstream benchmarks together (from reviewer 13xY). Work regarding the requested table and writing clarity were able to be incorporated quickly into the PDF revision to demonstrate the actual rewrites and table. We will post this revision asap (hopefully still visible to you if we miss the rebuttal period time frame). A sincere thank you for helping us improve our work!

---

> > ### Comment · Reviewer_QLxK · 2026-05-11
> >
> > Thank you for the additional results. It helped a lot to assess the usefulness of the proposed approach.

---

### Decision · Action_Editor_7dp1 · 2026-06-22

**Recommendation:** Accept with minor revision

**Additional Comments:**

The authors could incorporate the additional experiments during the rebuttal, such as those in response to Reviewer QLxK, into the final version.

**Audience:**

Yes

**Audience Explanation:**

The work is of broad interest to researchers working on model compression, efficient inference, and transformer architectures.

**Claims And Evidence:**

Yes

**Claims Explanation:**

This paper studies transformer model compression through the integration and merging of feed-forward sublayers. Although weight tying has been extensively studied in the literature, prior work has largely focused on training models from scratch. In contrast, this work investigates layer merging for pretrained models. Experimental results on both vision transformers and large language models demonstrate the effectiveness of the proposed approach and the main claim of the paper.

Several concerns were raised during the initial review process. During the rebuttal, the authors provided additional experiments that largely address these concerns and strengthen the support for their claims. The authors also indicated that these additional results, particularly those provided in response to Reviewer QLxK, including the comparison with quantization methods, will be incorporated into the final version.